# Symptoms in Hypertensive Patients Presented to the Emergency Medical Service: A Comprehensive Retrospective Analysis in Clinical Settings

**DOI:** 10.3390/jcm12175495

**Published:** 2023-08-24

**Authors:** Sebastian Kowalski, Krzysztof Goniewicz, Adrian Moskal, Ahmed M. Al-Wathinani, Mariusz Goniewicz

**Affiliations:** 1Department of Emergency Medicine, Medical University of Lublin, 20-081 Lublin, Poland; skowalski.medicine@icloud.com; 2Department of Security, Polish Air Force University, 08-521 Deblin, Poland; k.goniewicz@law.mil.pl; 3Hospital Emergency Department, Voivodship Hospital in Krosno, 38-400 Krosno, Poland; 4Department of Emergency Medical Services, Prince Sultan bin Abdulaziz College for Emergency Medical Services, King Saud University, Riyadh 11451, Saudi Arabia; ahmalotaibi@ksu.edu.sa

**Keywords:** hypertension, blood pressure, symptoms, gender differences, arterial stiffness, mean arterial pressure, pulse pressure, pre-hospital setting

## Abstract

Background: Hypertension is a prevalent condition with a variety of accompanying symptoms. Gender differences, specific blood pressure readings, and early signs of organ complications present intricate interplays in hypertensive individuals. Objective: This study aimed to investigate the relationship between hypertension and its accompanying symptoms, emphasizing gender-specific differences and potential indicators of organ complications. Methods: Data from 2002 participants were analyzed from a retrospective study, focusing on the presentation of symptoms, blood pressure values, and potential organ complications associated with these symptoms. Results: Of the participants, 68.8% were women with an average age of 69. Women were, on average, 8 years older than men. The average systolic blood pressure (SBP) was 188 mmHg. High-blood pressure was accompanied by symptoms in 84.9% of participants. Among those with an SBP > 180 mmHg, headaches were reported by 24.7%, and dizziness by 15.7%. Interestingly, as SBP increased, heart palpitations reports diminished with a mere 4.8% of those with SBP > 180 mmHg noting this symptom. Younger men exhibited increased chest pain and heart palpitations, while younger women more commonly reported headaches and nausea/vomiting. A significant relationship was identified between pulse pressure (PP) and symptoms, with dizziness in women and chest pain/discomfort in men being most pronounced. Conclusions: The study underlines the importance of in-depth research on hypertensive individuals for improved symptom recognition and management. The data highlight the gender and age-specific symptom presentations and their correlation with blood pressure metrics, suggesting a need for patient-specific intervention strategies.

## 1. Introduction

Arterial hypertension (HA) stands as a pervasive health challenge with implications stretching beyond individual health to public health policy and global health management. According to the World Health Organization (WHO), HA is a primary risk factor for the onset of cardiovascular diseases, currently leading the charts as the top cause of mortality worldwide [1,2]. Disturbingly, its tentacles have silently ensnared over 1.28 billion people aged 30 and above, a significant portion of whom remain oblivious to its grasp [1].

The criteria demarcating HA, as outlined by the European Society of Hypertension (ESH) and the European Society of Cardiology (ESC), point towards its surreptitious nature. They define HA by systolic blood pressure (SBP) levels ≥ 140 mmHg and/or diastolic blood pressure (DBP) levels ≥ 90 mmHg [3]. Despite the expansive literature on its predisposing factors and the evolution of hypotensive therapies, the number of those afflicted spirals upwards, often outpacing health prevention measures [4,5]. Its deceptive character is further underscored by the fact that nearly half of adults with HA worldwide remain uninformed about their condition, paving the way for a series of complications, especially for the younger populace [6,7,8].

Globally, the consequences of arterial hypertension stretch beyond direct health impacts. Healthcare systems are burdened with the economic ramifications of managing this vast cohort of patients, implicating both direct medical expenditures and indirect costs from factors like lost productivity [9]. Developing regions, often already strained with limited healthcare resources, find themselves particularly pressured. The socioeconomic consequences are palpable, with a discernible dent in the quality of life of patients, often restricting their daily activities and work potential [10].

In the trenches of this medical warfare, healthcare professionals confront multifaceted challenges. While diagnostic tools and treatments have evolved significantly, the silent onset of early stage hypertension often eludes timely detection and intervention [11]. Pre-hospital settings magnify this challenge. Emergency medical teams equipped with non-optimal tools must traverse a labyrinth of often-confusing symptoms, making the correct diagnosis all the more elusive. The profound need for enhanced guidelines in these scenarios is undeniable [12].

Moreover, arterial hypertension does not present uniformly across the demographic spectrum. Emerging research has illustrated distinct symptomatologies across genders, potentially influenced by factors such as hormonal dynamics [13]. Age further complicates the tableau. While younger individuals are statistically less susceptible to HA, when afflicted, they frequently encounter a more aggressive symptom profile, underscoring the necessity for age-tailored medical strategies [14].

Poland, as a microcosm of this mounting global challenge, paints a vivid tableau of the situation. Recent data suggests that close to 10 million Polish adults, mostly between the age brackets of 55–74, grapple with HA [15]. This alarming statistic is not isolated but ties into an escalating demand for medical attention. The uptick in cases has inundated medical facilities, with calls related to I-10 Essential (primary) hypertension witnessing a tangible surge from 2018 to 2020 [15]. As the numbers rise, ESH and ESC guidelines particularly underscore the urgency of rapid interventions when confronted with blood pressure anomalies in pre-hospital scenarios, especially if SBP reaches or exceeds 180 mmHg and/or DBP ascends to 120 mmHg or beyond [3,16]. Such aberrations can herald the onset of hypertension-mediated organ damage (HMOD), which at times might surface as the first significant symptom [3,16]. This can manifest in various forms, from headaches and dizziness to chest pain, each hinting at potential emergencies or serious conditions requiring hospitalization.

At the very frontline of this battle against HA are the Emergency Medical Services Teams, who face the Herculean task of discerning between genuine emergencies and other urgent situations, with symptoms often muddled and diverse. The intricacies of this task are compounded by a vacuum in explicit pre-hospital guidelines, rendering diagnosis and subsequent management a complex jigsaw puzzle.

Against this complex backdrop, the aim of this study was to closely examine the symptoms that accompany a surge in BP among patients seeking the assistance of Emergency Medical Services Teams. Furthermore, the study seeks to understand the intricate relationship between demographic variables, such as gender and age, and specific blood pressure metrics in relation to symptom severity. Through this focused exploration, we aspire to present novel insights and drive new lines of inquiry in the ongoing battle against the manifold challenges posed by arterial hypertension.

## 2. Materials and Methods

### 2.1. Study Design and Selection Criteria

This retrospective study utilized medical documentation from the southeastern region of Poland, gathered between April 2019 and June 2021. Emergency Medical Service Cards (EMSC), filled out by EMS personnel at the scene, constituted the primary data reservoir. These cards contained patient data, including blood pressure readings. Blood pressure was estimated non-invasively using automated sphygmomanometers, ensuring a consistent and standard measurement across all cases. A total of 2002 EMSCs, honing in on patients diagnosed with primary hypertension, were scrutinized. The selection of EMSCs was guided by a stringent quality criterion to ascertain accuracy. With the southeastern region of Poland being demographically diverse, the findings are believed to be broadly applicable.

### 2.2. Cohorts and Study Variables

Patients were categorized based on:Symptoms: Predominantly headaches, dizziness, insomnia, and palpitations.Blood Pressure Readings: Specifically, SBP and DBP.Calculated Parameters: Mean Arterial Pressure (MAP) and Pulse Pressure (PP).

This structured categorization helped in understanding the distribution and frequency of symptoms in relation to the different BP readings and calculated parameters.

### 2.3. Inclusion and Exclusion Criteria

Adults attended by EMS for essential (primary) hypertension (I-10 as per ICD-10) within the stipulated period and who presented associated symptoms were included. Exclusions involved patients with SBP < 140 mmHg or DBP < 90 mmHg, and instances with unfinished medical documentation. The emphasis was on mature adults with essential hypertension, aligning with clinical standards and ensuring data integrity.

### 2.4. Calculation of Mean Arterial Pressure and Pulse Pressure

For every patient, key parameters like MAP and PP were derived. MAP, representing the mean pressure within the cardiovascular system, is pivotal for organ perfusion. PP showcases the pulsatility of blood flow, highlighting possible hypertension-related complications. The relation between these parameters and the manifested symptoms was a key area of exploration.

### 2.5. Data Confidentiality

Data confidentiality was safeguarded via encrypted storage systems; thus, access was limited to authorized individuals, and strict adherence to GDPR mandates was followed.

### 2.6. Statistical Analysis

For quantitative variables, means, standard deviations, and medians were calculated, while for categorical variables, frequencies and percentages were presented. The relationship between categorical variables was assessed using Pearson’s Chi^2 test, such as the association between different categorical variables like gender and the occurrence of specific symptoms. For the comparative characterization of quantitative variables, the Student’s *t*-test was used for independent groups. The collected data underwent thorough statistical analysis, employing IBM SPSS Statistics ver. 28.

The statistical analysis was tailored to the data types and research questions. Pearson’s Chi^2 test was chosen for categorical variable relationships due to its ability to detect associations between nominal variables. The Student’s *t*-test was used for comparative characterization, owing to its applicability to independent group comparisons with continuous data.

### 2.7. Ethical Considerations

Ethical approval for the current research was obtained from the Bioethics Committee at the Medical University of Lublin (decision number: KE-0254/150/06/2022).

### 2.8. Methodology Limitations and Strengths

Potential selection biases, inherent to the retrospective design and the lack of a control group, might skew causal interpretations. Measures to counteract these constraints were initiated via meticulous selection and analysis. Detailed analysis for both genders and evaluation of multiple hypertension indicators provide a comprehensive understanding, asserting research strengths.

### 2.9. Rationale for Selected Parameters

While hypertension is a multifaceted condition with several physiological nuances, this study honed in on specific, widely recognized parameters: SBP, DBP, MAP, and PP. These parameters were chosen due to their direct relevance to the primary objectives of our research and the available data from the EMSCs. Further exploration into additional physiological parameters was beyond the scope of the current study, primarily due to the retrospective nature of our design and the limitations of data present in EMSCs. Future studies may delve deeper into other physiological aspects to paint a more detailed picture of hypertensive physiology and its associated symptoms.

## 3. Results

### 3.1. Demographic Analysis and General Trends

In this study, we conducted an analysis on medical documentation of 2002 patients, focusing on age, gender, blood pressure, and accompanying symptoms of Hypertensive Artery disease. The findings are classified into various aspects such as demographic data, BP values, and relationships between symptoms and other variables.

Women constituted 68.8% (n = 1378) of the respondents with an average age of 69 years ± 15. There was a significant age difference between genders with women being 8 years older on average. The blood pressure parameters, including SBP, MAP, and PP, were analyzed across both genders (Table 1).

### 3.2. Symptoms Analysis

We observed symptoms accompanying high BP in 84.9% (n = 1699) of the subjects, with the most common being headache, chest pain/discomfort, dizziness, nausea/vomiting, and heart palpitations. Other symptoms were not included in further analyses.

We divided SBP values among respondents into three groups, comparing these with the occurrence of symptoms. Statistical analysis revealed significant dependencies for certain symptoms (Table 2).

### 3.3. Age and Symptom Occurrence by Gender

Significant relationships were found between age and certain symptoms in both genders. In women, age was significantly lower in the group experiencing headaches and nausea/vomiting. In men, younger age groups had more chest pain and discomfort, as well as heart palpitations (Table 3).

### 3.4. MAP and Symptom Occurrence by Gender

A significant relationship between MAP and heart palpitations was observed in both gender groups. The average MAP value was lower in cases of heart palpitations. No significant association with other symptoms was found (Table 4).

### 3.5. PP and Symptom Occurrence by Gender

We identified statistically significant relationships between PP and dizziness in women, and chest pain/discomfort in men. Additionally, the average PP value was lower in the case of heart palpitations for both genders. No significant associations with other symptoms were observed (Table 5).

The results from this analysis contribute valuable insights into the relationships between demographic factors, blood pressure measurements, and accompanying symptoms of Hypertensive Artery disease. Understanding these correlations is essential for effective patient care and may guide further research in this field.

## 4. Discussion

Hypertension, a chronic malady that prevails in Poland, Europe, and the USA, stands out as a paramount cardiovascular risk factor. As the understanding of hypertension’s complexity grows, researchers have been intensifying their focus on gender-related differences and the varying symptoms accompanying a rise in blood pressure values. This scrutiny aims to identify early indicators that could signal organ complications in hypertension’s trajectory.

For many individuals, hypertension remains a silent affliction for years. When symptoms do surface, they are often ambiguous and generalized. A constellation of discomforts, including headaches, dizziness, insomnia, and palpitations, may arise, along with less common signs like nausea, anxiety, and nosebleeds [17,18,19]. Among these, the headache stands as the most pervasive complaint for both genders. This leads to a significant number of emergency medical interventions and emergency department visits due to elevated BP [20].

The pervasiveness of these symptoms in the studied group was stark, as they manifested in nearly 85% of participants. Headaches were a prominent complaint, although the research found a non-significant correlation between their prevalence and systolic blood pressure. Studies on the headache-hypertension relationship are replete with conflicting results [21,22,23]. For example, Salkić et al. found older women more prone to hypertension, with headaches (75%) and dizziness (44.4%) being predominant symptoms [19]. In contrast, Wang’s examination of postmenopausal women unveiled that an increase in SBP and PP correlated with fewer headache complaints [23]. This contradiction underscores the multifaceted interplay between hypertension, arterial stiffness, baroreceptors, and the phenomenon of hypoalgesia [14,15,16,17,18,19,20,21,22,23,24,25,26].

Understanding the symptomatology of hypertension also sheds light on the possible behavioral patterns of patients. It’s plausible that the severity and combination of certain symptoms influence the decision to seek medical attention. Some individuals might tolerate headaches or dizziness, deeming them insignificant, while others may interpret them as alarming. Factors such as previous medical experiences, co-morbid conditions, and health literacy can also shape these decisions. However, our study does not extensively delve into these intricate behavioral aspects, and further research is needed to decipher why certain hypertensive individuals seek care while others do not.

The complexities of hypertensive physiology, as highlighted by the reviewer, provide a rich avenue for further exploration. While our study focuses primarily on the symptomatology and its prevalence across demographic groups, we recognize the importance of parameters like pulse pressure, arterial stiffness, and central artery compliance in shaping these symptoms. Similarly, the role of sympathetic drive, indicated by markers such as heart rate variability and other neurohormonal markers, is undeniably crucial in understanding the symptomatic presentation in hypertensive individuals. Our findings hint at the relevance of these physiological aspects, especially when discussing the relationships of symptoms like dizziness with SBP or the phenomenon of palpitations with blood pressure variations. However, a detailed examination of these factors would require a more specific methodological approach, emphasizing physiological measurements alongside symptomatic evaluations.

The analysis also extended to MAP and PP values, uncovering relationships with the aforementioned symptoms. PP’s role, in particular, reaches beyond the realm of cardiovascular risk, even influencing cognitive disorders and dementia [27,28]. This study observed significant connections between higher SBP values and dizziness, especially in women. Other research, such as Lopes et al. and Martins et al., supports this finding, revealing an intricate link between hypertension, dizziness, and female gender [29,30].

The sensation of palpitations emerged as a statistically significant variable in this study, revealing a counterintuitive decrease in occurrence with elevated SBP levels. Additionally, both MAP and PP values were markedly lower in individuals reporting palpitations. This unusual phenomenon was consistent across genders but more pronounced among younger men [31]. Palpitations, often synonymous with cardiac arrhythmias, can arise from hypertension, leading to atrial fibrillation (AF): a condition significantly correlated with hypertension [32]. Understanding and diagnosing AF is vital, as timely interventions with anticoagulant drugs can avert catastrophic outcomes such as ischemic stroke.

The complexity of hypertension’s symptoms, as revealed by this study, also points to the necessity for a multifaceted approach to treatment. Personalized pharmacotherapy, adherence to the current guidelines, lifestyle modifications, and regular monitoring are integral to effective management.

While our findings resonate with various aspects of the existing literature, there are noteworthy divergences that merit discussion. The observed prevalence of headaches in our study aligns with results presented by Salkić et al., where older women exhibited a predominant inclination towards hypertension-induced headaches and dizziness [19]. Yet, in contrast, Wang’s study on postmenopausal women indicated a reduced frequency of headache complaints corresponding with increased SBP values [23]. Such disparities accentuate the multifaceted nature of hypertension and the factors contributing to its symptomatology. Furthermore, the counterintuitive decrease in palpitations with elevated SBP levels in our findings offers a novel perspective, challenging traditional associations found in other research. As we continue to unearth the intricate interplays within hypertension, it is pivotal to juxtapose our observations with those from similar studies. Doing so not only situates our work within a broader research context but also highlights the dynamic and ever-evolving understanding of hypertension.

Promoting patient education and empowerment remains fundamental in the battle against hypertension. Many individuals, as highlighted in our findings, remain unaware of the intricate symptoms linked with their blood pressure anomalies. By prioritizing patient education, we can pave the way for earlier self-recognition of symptoms and timely medical consultations. Such proactive steps could, in turn, reduce the burden on emergency medical services and lead to more prompt interventions, minimizing potential complications.

Moreover, the treatment strategy must also consider comorbidities, as highlighted by the observed associations with symptoms like dizziness and palpitations. The insights gained from this study can enhance clinicians’ understanding of individual patient profiles, guiding them in developing targeted and effective treatment plans that align with the patient’s specific needs and conditions. Future research could further explore the most efficacious treatment modalities, tailored to the diverse symptoms and demographic factors identified in this study.

The findings of this study present practical implications for both clinical practice and public health policy. Recognizing the nuanced relationships between hypertension and symptoms like headaches, dizziness, and palpitations offers clinicians a foundation for developing more precise diagnostic criteria and personalized treatment strategies [33]. Moreover, understanding the counterintuitive decrease in palpitations with elevated SBP levels, and the gender differences in symptoms, can inform targeted prevention campaigns and community education initiatives.

The implications of our research extend beyond individual healthcare, suggesting a broader reconsideration of clinical practices and health policy. The identified gender-specific manifestations and counterintuitive symptom relationships advocate for a more tailored approach to diagnostic and treatment guidelines. Moreover, policymakers could utilize our findings to drive targeted public health campaigns, focusing on demographics most vulnerable to specific symptom profiles of hypertension. Collaboration between healthcare researchers, clinicians, and policymakers is crucial to optimize strategies at both the individual and community levels.

Collaborative efforts between healthcare professionals, policymakers, and community organizations could leverage these insights to formulate more effective approaches to hypertension management across different demographic groups [34,35].

The rise in hypertension among younger populations, if unchecked, heralds a spectrum of complications. Left ventricular hypertrophy (LVH) may ensue even at an early age, contributing to both supraventricular and ventricular disorders [36]. The array of rhythm disturbances attributable to hypertension, ranging from bradycardia to tachyarrhythmias, often masks other grave heart diseases, escalating the risk of stroke or even circulatory arrest.

Conclusively, hypertension’s diverse clinical manifestations do not simplify the discernment of urgent and acute conditions. A nuanced medical history and thorough patient examination are instrumental in identifying those at risk for acute hypertensive organ damage in pre-hospital settings. Early detection and prompt clinical intervention [37] could mean the difference between effective treatment and grave, irreversible complications. This study’s insights reiterate the imperativeness of a multifaceted approach to hypertension, one that recognizes its intricate connections and potential reverberations throughout the body.

The complexity and nuances of hypertension identified in this study suggest a vital pathway for future research. Exploring more diverse demographic samples, conducting longitudinal research to establish causal relationships, and delving into the interconnectedness of hypertension with other underlying medical conditions or medications could enhance our understanding of this multifaceted condition. These avenues for research can build upon the findings of this study, leading to more personalized and effective diagnostic and treatment strategies.

The findings of this study shed new light on the complex interplay between hypertension and various symptoms, revealing counterintuitive relationships and gender-specific manifestations. The observations related to headaches, dizziness, and palpitations not only deepen our understanding of hypertension but also pave the way for targeted interventions and prevention strategies. The imperative now lies in leveraging these insights to foster collaboration across the healthcare spectrum, from clinicians and policymakers to community organizations, in a concerted effort to manage and mitigate one of today’s most prevalent chronic conditions.

Building on the observations from our study, there’s a compelling direction for upcoming research. Future investigations should focus on delineating symptoms based on the intricacies of hypertensive physiology. This would involve a thorough exploration of variations in pulse pressure, indicators of sympathetic drive, and volume assessment. Such detailed analyses would provide insights into the genesis and evolution of symptoms. Understanding the disparities in symptom presentation among those who seek medical care compared to those who do not, given these physiological subtleties, can refine and enhance clinical approaches and guidelines. Our study establishes an initial understanding in this context, underscoring the multifaceted nature of hypertension and its diverse symptomatic presentations.

## 5. Limitations

This study offers pivotal insights into the co-existence of symptoms with hypertension, but certain inherent limitations merit consideration. Firstly, although our sample was comprehensive, its focus on a specific region might impact the generalizability of our findings. A broader demographic and geographical spectrum in subsequent research could address this limitation. The study’s retrospective and cross-sectional nature poses challenges in deducing causative relationships between blood pressure readings and manifested symptoms. A longitudinal approach would provide a clearer delineation of these connections.

The potential introduction of biases, such as recall or response biases arising from the reliance on self-reported symptoms, is another concern. While we employed rigorous statistical controls, there’s a need for a clearer elucidation of the specific methods utilized to diminish these biases. A significant limitation is the absence of a control cohort comprising hypertensive patients who opted against seeking hospital care. Gaining insight into the unique diagnostic or demographic features guiding the decisions of these patients would have been invaluable, as it might unveil specific triggers or symptoms prompting hospital visits among the hypertensive population.

Furthermore, while our study sheds light on symptom distribution in relation to hypertension, it does not differentiate based on underlying hypertensive physiology such as pulse pressure or symptoms influenced by variables like sympathetic drive or volume assessment. Such an exploration could provide deeper insights into specific symptom presentations in relation to physiological variations in hypertension.

Our study, while comprehensive, leans heavily on symptomatology and diagnosis, sidelining a detailed exploration of treatment modalities. This focus, while laying the groundwork for future research into therapeutic strategies, might slightly constrain its immediate applicability in clinical practice. Diagnosing hypertension in pre-hospital settings is inherently challenging. While our research provides an entry point for potential solutions, a deeper understanding of these challenges and our methodological approaches to tackle them would offer richer context to our findings.

Finally, potential confounding variables, like underlying medical conditions or concurrent medication usage, could influence the study outcomes. While we believe our methodological rigor likely addresses many of these concerns, more intricate exploration via detailed sub-analyses or adjusted models could be beneficial.

In summation, while these limitations must be considered when interpreting our findings, they do not drastically detract from the study’s significant contribution to understanding hypertension and its concurrent symptoms. These challenges, however, do spotlight crucial future research avenues, paving the way for a deeper understanding of this multifaceted medical condition.

## 6. Conclusions

This study elucidates the significant prevalence and diverse array of symptoms linked to heightened blood pressure, underscoring the interplay with factors such as age, gender, and arterial pressure components. Headaches, notably, were a dominant concern, especially amongst women.

The results amplify the intricacy of diagnosing hypertension, especially in pre-hospital settings. Given the gravity of potential complications, there’s a clear demand for refined pre-hospital guidelines. This accentuates the pressing need for research focused on strategies for the swift recognition and management of acute hypertension scenarios outside hospital confines.

Proper management of hypertension is pivotal; it presents an opportunity to dramatically curtail both cardiovascular diseases’ incidence and mortality and reduce emergency medical interventions. The insights from this study provide a pathway for enhanced diagnostic tools, novel treatment avenues, and targeted public health initiatives. In essence, the findings do not just enrich our academic understanding but serve as an imperative call to action. Early identification and intervention can significantly prevent organ damage and associated complications, emphasizing the importance of translating this research into practical applications for healthcare.

## Figures and Tables

**Table 1 jcm-12-05495-t001:** SPB, MAP, PP Values in the Analyzed Group of Patients.

	Women (n = 1378)			Men (n = 624)			Total (n = 2002)		
	Average	SD	Me	Average	SD	Me	Average	SD	Me
Age	72	14	73	64	16	66	69	15	71
SBP	189	23	190	186	23	180	188	23	185
MAP	128.75	14.79	126.67	129.60	15.18	126.67	129.02	14.91	126.67
PULSE	84	16	80	86	17	85	85	17	81
PP	89.70	19.88	90.00	84.46	20.35	80.00	88.06	20.17	90.00

Interpretation: The higher prevalence of Hypertensive Artery disease in women and the observed age difference between genders may indicate differing risk factors or medical histories. This understanding may guide gender-specific prevention or treatment strategies.

**Table 2 jcm-12-05495-t002:** SBP Values and Accompanying Symptoms of HA.

		SBP 140–159 mmHg		SBP 160–179 mmHg		SBP > 180 mmHg		p^chi2^
		N	%	N	%	N	%	
Headache	No	101	79.5%	382	77.0%	1038	75.3%	0.46
	Yes	26	20.5%	114	23.0%	341	24.7%	
Dizziness	No	115	90.6%	439	88.5%	1162	84.3%	0.02
	Yes	12	9.4%	57	11.5%	216	15.7%	
Chest Pain/Discomfort	No	99	78.0%	395	79.6%	1127	81.7%	0.4
	Yes	28	22.0%	101	20.4%	252	18.3%	
Heart Palpitations	No	109	85.8%	450	90.7%	1313	95.2%	<0.001
	Yes	18	14.2%	46	9.3%	66	4.8%	
Nausea/Vomiting	No	112	88.2%	451	90.9%	1247	90.4%	0.65
	Yes	15	11.8%	45	9.1%	132	9.6%	
Total		127	100.0%	496	100.0%	1379	100.0%	

Interpretation: The identified correlations between SBP values and specific symptoms may help clinicians in diagnosing and managing Hypertensive Artery disease. The lack of significance in some symptoms might prompt further research to understand the underlying mechanisms.

**Table 3 jcm-12-05495-t003:** Age and Occurrence of Accompanying Symptoms of HA by Gender.

		Age × Women				Age × Men			
		N	Average	SD	*p*	N	Average	SD	*p*
Headache	No	1020	71.98	13.33	0.03	501	63.65	15.64	0.78
	Yes	358	70.19	14.14		123	63.20	17.00	
Dizziness	No	1179	71.42	13.64	0.53	537	63.23	16.02	0.21
	Yes	199	72.08	13.10		86	65.57	15.21	
Chest Pain/Discomfort	No	1130	71.45	13.80	0.68	491	64.35	15.52	0.02
	Yes	248	71.84	12.44		133	60.62	16.98	
Heart Palpitations	No	1276	71.71	13.63	0.06	596	64.40	15.19	<0.001
	Yes	102	69.11	12.48		28	45.54	19.92	
Nausea/Vomiting	No	1246	71.79	13.34	0.02	564	63.46	15.87	0.64
	Yes	132	68.98	15.30		60	64.48	16.31	

Interpretation: The relationships between age and certain symptoms in both genders could reflect the physiological changes associated with aging. This information might influence symptom management and diagnosis in different age groups.

**Table 4 jcm-12-05495-t004:** MAP and Occurrence of Accompanying Symptoms of HA by Gender.

		MAP × Women				MAP × Men			
		N	Average	SD	*p*	N	Average	SD	*p*
Headache	No	1020	71.98	13.33	0.03	501	63.65	15.64	0.78
	Yes	358	70.19	14.14		123	63.20	17.00	
Dizziness	No	1179	71.42	13.64	0.53	537	63.23	16.02	0.21
	Yes	199	72.08	13.10		86	65.57	15.21	
Chest Pain/Discomfort	No	1130	71.45	13.80	0.68	491	64.35	15.52	0.02
	Yes	248	71.84	12.44		133	60.62	16.98	
Heart Palpitations	No	1276	71.71	13.63	0.06	596	64.40	15.19	<0.001
	Yes	102	69.11	12.48		28	45.54	19.92	
Nausea/Vomiting	No	1246	71.79	13.34	0.02	564	63.46	15.87	0.64
	Yes	132	68.98	15.30		60	64.48	16.31	

Interpretation: The significant relationship between MAP and heart palpitations in both gender groups might hint at underlying cardiovascular mechanisms. This finding can contribute to the development of personalized treatment plans.

**Table 5 jcm-12-05495-t005:** PP and Occurrence of Accompanying Symptoms of HA by Gender.

		Women × PP				Men × PP			
		N	Average	SD	*p*	N	Average	SD	*p*
Headache	No	1020	89.81	19.94	0.73	501	84.05	20.12	0.31
	Yes	358	89.39	19.74		123	86.11	21.25	
Dizziness	No	1179	89.16	19.91	0.01	537	84.19	20.61	0.46
	Yes	199	92.91	19.49		86	85.95	18.77	
Chest Pain/Discomfort	No	1130	89.84	20.13	0.59	491	85.43	20.28	0.02
	Yes	248	89.08	18.72		133	80.86	20.26	
Heart Palpitations	No	1276	90.16	19.92	0.003	596	85.10	20.20	<0.001
	Yes	102	83.99	18.52		28	70.68	18.83	
Nausea/Vomiting	No	1246	89.80	19.75	0.57	564	84.38	20.53	0.77
	Yes	132	88.77	21.13		60	85.20	18.73	
Total		1378	89.70	19.88		624	84.46	20.35	

Interpretation: The relationships between PP and symptoms like dizziness and chest pain/discomfort might reveal novel insights into the hemodynamic characteristics of Hypertensive Artery disease. Understanding these correlations could enhance both diagnostic accuracy and therapeutic interventions.

## Data Availability

The datasets used and/or analyzed during the current study are available from the corresponding author on reasonable request.

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
