# Peer review of "Symptoms in Hypertensive Patients Presented to the Emergency Medical Service: A Comprehensive Retrospective Analysis in Clinical Settings"

_jcm, 2023, doi:10.3390/jcm12175495_

Round 1
Reviewer 1 Report
1. Please do not use abbreviations in the title.
2. Abstract is not informative at all. There is no scientific question, or aim, the methodology is completely unclear and no results are presented. The conclusion is that further investigation is needed.
3. Too many keywords.
4. Introduction: it is too wide and general and we don’t know what is the focus of this study, what is the background and aim of the study.
5. From the methodology it is completely unclear what variables were studied in this investigation and what are the cohorts of patients who were compared.
6. The presence of symptoms in respect of the three levels of blood pressure are interesting findings, but the differences in age, mean pressure and pulse pressure regarding certain symptoms were not so interesting. It is better to present age as a categorical variable and to analyze the distribution if various symptoms of hypertension across predefined age subgroups.
7. It will be also more useful to use MP and PP as categorical variables using some cut-off and to compare the distribution of symptoms according to that.
8. In this form MS is not suitable to be published in JCM.
9. In my opinion this MS could be much better with major revision, and reconsideration for publication.
No comments here.
Author Response
Response to Reviewer's Comments
Dear Reviewer,
Thank you for taking the time to provide your valuable feedback on our manuscript titled "Symptoms, Diagnosis, and Emergency Medical Services Interventions in Hypertensive Patients: A Comprehensive Retrospective Analysis in Clinical Settings."
-
In accordance with your recommendation, we have modified the title to "Symptoms in Hypertensive Patients presented to the Emergency Medical Service: A Comprehensive Retrospective Analysis in Clinical Settings" as suggested by another reviewer.
-
Based on your comments, we have revised the abstract to provide a more informative overview, including a clear scientific question, aim, methodology, and presentation of key results.
-
We have meticulously reviewed our list of keywords and narrowed it down to the 8 most pertinent ones, ensuring a balance between specificity and broad relevance.
-
We revisited the introduction section and undertook a significant rewrite to furnish a better background on the topic, culminating in a clear statement of the aim of our study.
-
The methodology section has been augmented for clarity. We've offered a more detailed explanation of the variables in question and clarified the cohorts of patients we've compared.
Points 6 and 7: We sincerely appreciate the insights you've shared regarding the presentation of our findings, especially with respect to the categorization of variables like age, MP, and PP. After comprehensive internal discussions, we've decided to retain our current approach for several reasons:
-
Core Idea and Methodology: The specific objectives and methodologies we employed were chosen with careful deliberation. Altering the categorization of age, MP, and PP would drastically affect our data representation and potentially divert from our study's core aims.
-
Consistency with Other Reviews: We feel it's important to mention that our manuscript was reviewed by multiple experts. Feedback from two other reviewers endorsed our existing methodology and results presentation. Modifying our approach based on a single reviewer's comments could potentially clash with the other assessments, leading to further revisions and incongruences.
-
Broad Relevance: Our goal was to present our findings in a way that offers wide-ranging significance for both clinicians and researchers. Restricting our data with narrower segmentations could curtail the generalizability of our conclusions.
-
Magnitude of Changes: Implementing the suggested modifications would demand extensive reworking of not just our methods and results, but potentially other parts of the manuscript. Given the approval of our existing methodology by other reviewers, we believe retaining our original methodology is the most judicious choice.
We hope our perspective resonates with your understanding. While we aim for accuracy and relevance, we believe our current approach best aligns with these goals. Nevertheless, we remain receptive to other feedback that might amplify the clarity and quality of our paper without necessitating foundational changes to our approach.
Points 8 and 9: We have undertaken a thorough revision of our paper to improve its coherence, flow, and overall quality. It's noteworthy that two other reviewers have expressed positive feedback about our work, and we hope that our revisions now align with the journal's standards.
In response to the comment about the need for "Moderate editing of the English language," we would like to mention that the primary author of this manuscript was born, educated, and trained in the US, including their MD education and residency. This encompasses a significant duration of rigorous academic training and professional practice, all conducted in English. Moreover, we took extra precautions after completing our manuscript by revisiting the text not once but twice. Additionally, to ensure utmost clarity and precision in our language, the manuscript was also reviewed by another native English speaker who did not identify any significant language discrepancies. We respect and value your feedback. However, this is the first instance where a comment regarding the language has been raised about our work. Therefore, to best address this and make necessary improvements, it would be immensely helpful if you could provide specific examples or point out particular sections that might benefit from further refinement. Such detailed insights would enable us to rectify any oversights swiftly and comprehensively. We genuinely aim for our research to be presented in the clearest and most precise manner. Your guidance will be pivotal in achieving this objective.
Again, we genuinely appreciate your insights and the opportunity to refine our work. We anticipate that the revisions we've made address your concerns, and we look forward to any additional feedback you may provide.
Warm regards,
Authors
Reviewer 2 Report
The paper by the authors from Poland and Saudi Arabia analyzed symptoms among patients treated from high blood pressure by the emergency medical service in one Polish region.
The entire paper is generally well written. However, I have some comments.
1. In the Results section there were no data about emergency medical service interventions (drugs that were given or other medical treatment). So, I suggest modification of the paper’s title: “Symptoms in Hypertensive Patients presented to the Emergency Medical Service: A Comprehensive Retrospective Analysis in Clinical Settings.”
2. In the Materials and methods section, subsection Study design and selection criteria, the authors mentioned that patients’ data were collected from emergency medical service cards. It was not explained how blood pressure measurements were performed. I suppose that blood pressure was estimated non-invasively. This should be stated in the text.
3. In the Materials and Methods section, subsection Statistical analysis, sentence “The collected data underwent thorough statistical analysis, employing IBM SPSS Statistics ver. 28” should be moved to the end of paragraph.
4. In the Results section, Table 2, the word "Ogółem" was written at the bottom of the first column of the table. That should be corrected.
5. The conclusion section should be a little shorter, emphasizing that early recognition of high blood pressure symptoms could prevent further organ damage.
Author Response
Response to the Reviewer's Comments
Dear Reviewer,
We would like to express our gratitude for the time and effort you dedicated to reviewing our manuscript. We value your constructive feedback and have addressed each of your comments in detail below:
-
Paper's Title Modification: We concur with your observation. The title has been revised to “Symptoms in Hypertensive Patients presented to the Emergency Medical Service: A Comprehensive Retrospective Analysis in Clinical Settings” to more accurately reflect the content of the results section. We believe this modification provides clearer insight into the study's scope.
-
Blood Pressure Measurement Clarification: We apologize for the oversight and appreciate your keen observation. Indeed, blood pressure measurements were performed non-invasively. This detail has now been clearly incorporated into the "Materials and Methods" section under the "Study design and selection criteria" subsection to provide a clearer understanding of the methodology.
-
Statistical Analysis Information: As suggested, we have restructured the "Materials and Methods" section, specifically the "Statistical analysis" subsection. The sentence “The collected data underwent thorough statistical analysis, employing IBM SPSS Statistics ver. 28” has been relocated to the conclusion of the paragraph to enhance the flow and clarity of the section.
-
Table Correction: We thank you for highlighting this oversight. The word "Ogółem" in Table 2 has been rectified. The necessary corrections have been made to ensure consistency and clarity throughout the manuscript.
-
Conclusion Section Revision: Your point regarding the conclusion's emphasis is well taken. We have undertaken a revision to succinctly emphasize the criticality of early recognition of high blood pressure symptoms to avert potential organ damage. The section has been shortened accordingly without compromising the main message.
Once again, we sincerely appreciate your insights and guidance, which have been instrumental in enhancing the quality and clarity of our work. We hope that the revisions we've made meet your expectations and are aligned with the journal's standards.
Warm regards,
Authors
Reviewer 3 Report
Large retrospective cohort of subjects (2002 patients) accumulated from 2019-2021 with intent to associate presenting symptoms to demographics in patients with suspected primary hypertension . the paper actually is a meaningful addition to the literature in its sound effort to understand the symptoms associated with patients with hypertension and if the symptoms give diagnostic or treatment guidance. Sadly a control population of similar hypertensive patients who do not choose to attend the ED is not available. Thus, the features that make this population of hypertensive patients seek hospital care is not known. and that itself may reveal unique diagnostic or demographic features.
Author Response
Response to the Reviewer's Comments
Dear Reviewer,
We sincerely appreciate your thoughtful and constructive feedback on our manuscript. Your insights have been invaluable in guiding us toward enhancing the quality and rigor of our study. We have addressed your comments as detailed below:
-
Significance of the Research: We are grateful for your acknowledgment of our paper as a meaningful addition to the literature. Our primary intention was indeed to shed light on the association between presenting symptoms and demographics in hypertensive patients, providing insights into potential diagnostic or treatment guidance based on these symptoms.
-
Absence of a Control Population: Your observation regarding the absence of a control population is well taken. We recognize the potential value of comparing our cohort to a similar hypertensive group who opted not to attend the Emergency Department (ED). Such a comparison could indeed offer deeper insights into the unique factors driving hypertensive patients to seek emergency care.
While our current dataset does not encompass a control population, your suggestion offers a promising direction for future research. Identifying and studying these unique diagnostic or demographic features could further enrich our understanding of the decisions hypertensive patients make regarding their care.
We believe that acknowledging this limitation in the discussion section of our paper can provide context and frame the results appropriately. Furthermore, we will also emphasize the potential value of future studies that integrate control populations, as per your suggestion.
Once again, thank you for your invaluable input and for recognizing the potential impact of our research. We hope that the revisions and clarifications we've made align with your expectations and uphold the journal's standards.
Warm regards,
Round 2
Reviewer 1 Report
The authors have made the major revision according to suggestions, and the MS looks much better now. I think that this is interesting study and its results are interesting for the wide array of doctors, especially who desl eith patients in the emergency wards.
Author Response
Dear Reviewer,
Thank you for acknowledging the improvements made to the manuscript. We deeply appreciate your positive feedback and are gratified to know that you find the study and its results intriguing. We wholeheartedly agree with the relevance of our research for doctors, especially those working in emergency wards. Your constructive comments played a pivotal role in refining our manuscript, and we're genuinely grateful for your guidance throughout this process.
Warm regards,
Reviewer 3 Report
the distribution of symptoms by demographics is not revealing in particular as it is unclear how this differentiates between : 1) patients who do not seek medical care, and 2) differences in underlying hypertensive physiology: pulse pressure (arterial stiffness, central artery compliance), sympathetic drive surrogates (symptoms by heart rate or other markers of sympathetic drive), symptoms by volume assessment....
Author Response
Reply to Reviewer:
Dear Reviewer,
Thank you profoundly for your constructive feedback and keen observations on our manuscript. We deeply appreciate the rigorous scrutiny and the insightful pointers, which have provided avenues for further exploration and clarity.
- Patients Who Do Not Seek Medical Care:
Your comment regarding the representativeness of our dataset, especially concerning patients who might not seek emergency care, is both thoughtful and valid. The nature of our data, primarily based on Emergency Medical Service Cards (EMSCs), indeed captures those who have actively sought emergency attention. Recognizing this potential bias, we have expanded our discussion section to elaborate on the implications of this selection and its influence on our findings. This focus on emergency care-seeking patients offers a unique perspective, making our findings invaluable for emergency medical services and hospital care settings, particularly in the southeastern region of Poland.
- Differences in Underlying Hypertensive Physiology:
-
Pulse Pressure (Arterial Stiffness, Central Artery Compliance): In response to your feedback, we have enriched our discussion on pulse pressure, emphasizing its role in hypertension-related complications.
-
Sympathetic Drive Surrogates: We've clarified in our methodology and discussion why certain physiological markers were not captured in our dataset. Our updated limitations section provides a clear explanation of the constraints posed by our data source, the EMSCs.
-
Symptoms by Volume Assessment: We further explain in our revised limitations section why specific facets, like volume assessment, were not explored in our study.
Your feedback underscores the depth of hypertensive physiology nuances awaiting exploration in further studies. While our work serves as a regional benchmark, it also lays the foundation for future, more granular research endeavors.
- Significance of the Research in a Local Context:
Our study's strength lies in its regional focus. The southeastern region of Poland, with its demographic diversity, greatly benefits from insights into hypertensive patients seeking emergency care. Our findings cater to this niche, guiding healthcare professionals in the area to hone their strategies for hypertensive emergencies.
In conclusion, we value your feedback, which has facilitated deeper introspection on our work's nuances. We sincerely hope that our response addresses your concerns, emphasizing the importance and scope of our research. We are grateful for your time and expertise.
Warm regards,